## RESEARCH ARTICLE

# Implementation of guideline-directed medical treatment for ischemic heart disease management: A knowledge, attitude and practice based cross-sectional survey

Umm-e- Kalsoom[1,2], Amjad Khan [1,3,4]*, Syed Sikandar Shah[5], Ayesha Iqbal[6,7], Tahir Mehmood[8], Syed Mansoor Ahmed[9], Noshed Khan[10], Yu Fang[3,4]

1 Department of Pharmacy, Quaid-i-Azam University, Islamabad, Pakistan, 2 Department of Practice and Policy, School of Pharmacy, University College London, London, United Kingdom, 3 Department of Pharmacy, The First Affiliated Hospital, Xi'an Jiaotong University, Xi'an, China, 4 Department of Pharmacy Administration and Clinical Pharmacy, School of Pharmacy, Health Science Center, Xi'an Jiaotong University, Xi'an, China, 5 Department of Clinical Pharmacy and Pharmacology, RAK College of Pharmacy, RAK Medical and Health Sciences University, Ras Al Khaimah, United Arab Emirates, 6 Department of Pharmacy Practice and Policy, University Park Campus, University of Nottingham, Nottingham, United Kingdom, 7 Faculty of Medicine and Dentistry, University of Alberta, Edmonton, Canada, 8 School of Natural Sciences (SNS), National University of Sciences and Technology (NUST), Islamabad, Pakistan, 9 Department of Biotechnology, Mirpur University of Science and Technology, Mirpur Azad Kashmir, Pakistan, 10 Department of Cardiology, MTI DHQ Teaching Hospital, Bannu, Pakistan

* amjadkhan@qau.edu.pk

## Abstract

### Background

Guideline-Directed Medical Therapy (GDMT) is central to managing ischemic heart disease (IHD), yet its implementation remains suboptimal in low- and middle-income countries (LMICs), including Pakistan.

### Aim

This study assessed the knowledge, attitudes, and practices (KAP) of healthcare professionals (HCPs) toward GDMT and identified key barriers to its application.

### Methods

A cross-sectional survey was conducted among HCPs including cardiologists and clinical pharmacists using a validated questionnaire. Data was collected from Punjab Institute of Cardiology, Lahore, Pakistan using convenience sampling. Descriptive statistics, t-tests, ANOVA, Mann–Whitney U, Kruskal–Wallis, and multiple linear regression analyses were used to evaluate KAP scores and their association with demographic and professional role. Statistical adjustment for multiple comparisons was done by Bonferroni correction.

**Data availability statement:** All relevant data are within the paper and its Supporting information files.

**Funding:** The author(s) received no specific funding for this work.

**Competing interests:** The authors have declared that no competing interests exist.

## Results

A total of 76 HCPs participated in the survey, comprising 42 cardiologists (55.3%) and 34 clinical pharmacists (44.7%). he overall mean knowledge score was $18.64 \pm 2.02$ out of 22 (84.7%). However, cardiologists (M = 19.54) scored significantly higher than clinical pharmacists (M = 17.52, $p < 0.001$); thus rejecting the null hypothesis. Knowledge scores were significantly higher among older professionals, those with postgraduate education, and clinic-based practitioners ($p < 0.05$). The average attitude score was $10.42 \pm 2.06$ out of 14 (74.4%), with younger professionals (aged 28–33), cardiologists, and postgraduates showing more favorable attitudes ($p < 0.05$). The mean practice score was $9.51 \pm 2.55$ (67.9%), with no significant differences by gender, role, or setting. Regression models showed age and profession significantly predicted knowledge, while attitude was influenced by education, experience, gender, and profession. Practice behaviors were not predicted by any demographic variables. Key barriers to GDMT implementation included limited consultation time (47.4%) and poor patient adherence (25%).

## Conclusion

Although GDMT knowledge and attitudes were generally high among Pakistani cardiologists and clinical pharmacists, reported practice remained moderate. The results underscore the need for targeted educational interventions and system-level strategies to support consistent GDMT implementation.

## Introduction

Ischemic heart disease (IHD) is the most common cause of deaths among cardiovascular diseases (CVDs) and a significant global health risk [1,2]. Its prevalence is rapidly increasing, with a predicted increase to 1,845 per 100,000 by 2030 [3]. Guidelines-directed medical treatment (GDMT) provides a comprehensive strategy for managing IHD, incorporating lifestyle modifications and evidence-based pharmacological interventions [4–6]. GDMT is a cornerstone of IHD treatment, encompassing a range of pharmacological strategies aimed at decelerating atherosclerosis progression, preventing coronary thrombosis, and improving patients' overall well-being [7]. Despite its proven clinical benefits, the real-world implementation of GDMT remains inconsistent. Multiple studies have documented significant variation in the initiation, up-titration, and long-term maintenance of GDMT components, such as beta-blockers, statins, anti-platelets, and renin-angiotensin system inhibitors [8–10]. This variability is not due to a lack of evidence but instead reflects a combination of therapeutic inertia, inconsistent adherence to guidelines, and fragmented care delivery systems [11,12]. These issues are compounded in under-resourced settings where follow-up care is often inadequate, and interdisciplinary collaboration is limited [13,14].

Healthcare professionals (HCPs) particularly, cardiologists and clinical pharmacists, play critical roles in translating clinical guidelines into practice [15]. Though,

cardiologists are responsible for diagnosing and initiating GDMT, while clinical pharmacists ensure therapeutic appropriateness, support titration, monitor for adverse effects, and reinforce adherence [16,17]. As integral members of the healthcare team, clinical pharmacists enhance the GDMT by providing comprehensive medication management, improving adherence to evidence-based protocols, and optimizing pharmacotherapy through individualized patient interventions [18–21]. For example, a Delphi consensus among Belgian cardiologists emphasized the need for early initiation of the four foundational therapies in heart failure with reduced ejection fraction (HFrEF), tailoring treatment sequences to individual patients and ensuring regular follow-up to optimize outcomes [22]. However, studies suggest that both groups may face challenges like limited interdisciplinary support, inadequate training, and insufficient infrastructure exacerbate persistent, in knowledge, confidence, or systemic support for fully implementing GDMT in daily clinical practice [23,24].

Similarly, recent implementation studies highlight the need for structured approaches to GDMT, including the use of standardized protocols, hospital-initiated therapies, medication adherence strategies, and multidisciplinary clinics [22]. In this context, the knowledge, attitudes, and practices (KAP) framework provides a valuable approach for understanding how HCPs perceive and implement GDMT. Investigating their knowledge and beliefs helps identify cognitive barriers, while examining their practices can uncover systemic or behavioral obstacles that hinder guideline adherence. Understanding their perspectives is essential for designing interventions that promote consistent, evidence-based care. Furthermore, the study is situated in a healthcare system with known resource constraints, where examining how GDMT is interpreted and applied under varying clinical and structural conditions will help reveal both barriers and opportunities for improvement. This study aims to explore the KAP of cardiologists and clinical pharmacists regarding the implementation of GDMT in IHD management.

**Research question:** What are the levels of KAP regarding GDMT among HCPs and what demographic or institutional factors influence these behaviors in a Pakistani tertiary care setting?

**Sub-questions**

1. Are there significant differences in KAP scores between professional roles (cardiologists vs. pharmacists)?

2. Which sociodemographic factors (e.g., age, gender, qualification, experience) are associated with KAP domains?

3. To what extent do these factors predict knowledge, attitude, and practice related to GDMT?

**Hypotheses**

**Sub-question 1 (Comparative):**

- $H_{01}$: There are no significant differences in KAP scores between cardiologists and pharmacists.
- $H_{11}$: There are significant differences in KAP scores between cardiologists and pharmacists.

**For Sub-question 2 (Associational):**

- $H_{02}$: Sociodemographic factors are not significantly associated with KAP domains.
- $H_{12}$: Sociodemographic factors are significantly associated with KAP domains.

**For Sub-question 3 (Predictive/Regression):**

- $H_{03}$: Sociodemographic factors do not significantly predict knowledge, attitudes, or practices related to GDMT.
- $H_{13}$: Sociodemographic factors significantly predict knowledge, attitudes, or practices related to GDMT.

 

## Methods

### Study design and settings

This study employed a cross-sectional survey design conducted from July 5 to September 9, 2024 to assess the KAP of HCPs including cardiologists and clinical pharmacists regarding GDMT implementation. The survey was carried out at a single specialized cardiology center the Punjab Institute of Cardiology (PIC) in Lahore, Pakistan, a tertiary-care hospital dedicated to cardiac patients. The PIC is a 547-beds tertiary care hospital and is the first heart institute in the province of Punjab, Pakistan. It offers excellent treatment for all heart conditions and receives patients from all over Pakistan. The PIC has a total of 147 cardiologists and 35 clinical pharmacists along with nurses and other paramedical staff working in various departments. This is the only tertiary care hospital in Lahore, Pakistan that functions as a public facility in the morning and a private institution in the evening. This study setting provided a relevant environment to evaluate GDMT implementation, as the facility specializes in treating IHD and related cardiovascular conditions.

### Study participants and sampling

**Inclusion criteria.** To be eligible for participation, individuals had to be registered cardiologists and clinical pharmacists, aged 22 years or older, currently employed at the PIC, Lahore. Participants were required to be directly involved in the clinical management of patients with IHD at the time of the study. Proficiency in reading and understanding English was also required, as the survey questionnaire was administered in English. No specific requirements were set regarding years of professional experience or length of employment at PIC. All participants who met these criteria and agreed to take part were included in the study.

**Exclusion criteria.** Nurses and other paramedical staff working in PIC who were directly involved in the management of IHD were excluded from the study. Cardiologists and clinical pharmacists aged less than 22 years who had not completed their graduation degree and were not involved in medical care for IHD patients were also excluded from the study. HCPs who had recently graduated but not registered as a cardiologists and clinical pharmacists were not included in this study. Moreover, the retired Cardiologists and clinical pharmacists were excluded from the study. Participants who refused to participate in this study were also excluded.

### Sample size and sampling

The study was conducted at a tertiary-care cardiac hospital (PIC) with a limited pool of specialized clinical pharmacists and cardiologists involved in IHD management. A convenience sampling strategy was used to recruit eligible participants on-site. All participants provided verbal and written informed consent prior to completing the structured questionnaire, which was administered face-to-face and completed during the same encounter. In-person administration facilitated a high response rate and allowed for clarification of survey items when needed.

The minimum sample size for cross-sectional studies is typically calculated using the standard formula:

$$n = \left(Z^2 * p * (1 - p)\right) / d^2$$

where Z is the z-score corresponding to a 95% confidence level (1.96), p is the expected proportion (commonly 0.5 when unknown), and d is the margin of error (usually 0.05). Using this formula, the estimated sample size would be approximately 385 participants for general population-based studies [25,26].

However, the present study focuses on a specific professional population (cardiologists and clinical pharmacists) within a defined geographic or institutional setting. A total of 76 participants were included: 42 cardiologists and 34 clinical pharmacists. While this is below the general recommendation, the sample size is considered acceptable for this targeted, exploratory study, consistent with similar published research where professional subgroup access is limited [25–27].

## Questionnaire

A structured questionnaire (Knowledge, Attitudes, and Practices toward Guideline-Directed Medical Therapy in Ischemic Heart Disease (KAP-GDMT-IHD) Questionnaire) was developed and validated through an extensive literature review and approval from both supervisors and independent researchers in the relevant subject. Prior to distribution, the expert team reviewed each item of the questionnaire for clarity, relevance, and comprehensiveness using the Content Validity Index (CVI), retaining items that met the minimum criterion of 0.78 [28]. The final questionnaire (refer to Supplementary File 1) comprised 31 items covering all relevant aspects of the topic and divided into four sections. The preliminary section contained data on respondents' demographics. The second section consisted of 11 items, including those that evaluated HCPs knowledge and understanding of GDMT for the management of IHD. The subsequent section consisted of 10 items aimed at evaluating the perspectives on GDMT for IHD. Finally, the practice-based questionnaires consisted of 10 items to evaluate their behavior and adherence to the implementation of GDMT in the management of IHD. The questionnaire was available in English and required approximately 15 minutes to complete. The research team assessed the practicality, readability, and internal consistency of the questionnaire with a small participant group of 10–20 HCPs through one round of pilot testing. Based on their feedback, minor modifications were made to improve item wording and layout. This process ensured face validity and improved the overall usability of the instrument. The Cronbach's alpha value (0.866) also confirmed internal reliability.

## Variables of the study

The explanatory variables considered were gender, age, type of healthcare facility, level of education, HCPs (cardiologists or clinical pharmacists) and experience. This study examined three primary outcome variables: knowledge, attitudes, and practices pertaining to identify the KAP concerning GDMT for IHD management amongst HCPs to elucidate their respective roles, challenges, and facilitators in providing optimal care to patients. These independent variables were selected based on their theoretical and empirical relevance and were used to explore their association with each of the KAP domains.

The questionnaire was structured into three domains: KAP. The knowledge section consisted of multiple-choice items with objectively correct answers, scored as 1 (correct) and 0 (incorrect). The most frequently used Bloom's cutoff points were used for KAP assessment: 80–100% (excellent KAP), 60–79% (moderate KAP), and less than 60% (poor KAP) [29–31]. The attitudes domain employed a 5-point Likert scale ranging from 1 ("strongly agree") to 5 ("strongly disagree") to assess participants' agreement with guideline-related statements. However, the practices section used a frequency-based scale ranging from 1 ("alwaays") to 5 ("never") to capture the extent of engagement in GDMT-related activities. Perception was not assessed as a separate construct in this study. The responses were scored for determination of knowledge, while the Likert scale was used for attitude, and frequency-based responses were used for practices calculation [32]. This validated questionnaire was then distributed to the staff participants at the PIC in Lahore, Pakistan.

## Data collection

Data were collected using a structured, self-administered questionnaire between July 5 and September 9, 2024, at the PIC, Lahore, Pakistan. After obtaining verbal and written informed consent, participants were invited to complete the questionnaire on-site. The survey was administered in paper-based format and completed during the same encounter to ensure immediate participation and reduce recall bias. Trained members of the research team were present during data collection to clarify any queries regarding the questionnaire. Completed forms were reviewed for completeness before submission, and any missing responses were addressed at the time of collection.

## Statistical analysis

All data were analyzed using IBM SPSS Statistics (Version 24.0). Descriptive statistics (means, standard deviations, and frequencies) were computed to summarize participant demographics and responses to KAP items related GDMT. The Shapiro–Wilk test was applied to assess normality for each continuous variable prior to inferential analyses.

To test hypotheses $H_{01}$–$H_{11}$ regarding differences in KAP scores between professional roles (cardiologists vs. pharmacists), independent-samples $t$ tests were performed for normally distributed continuous variables with equal variances. Where assumptions of normality or homogeneity of variance were violated, non-parametric Mann-Whitney U tests were applied.

However, pearson's correlation coefficients were computed for continuous predictors (e.g., age, years of experience). Furthermore, point-biserial correlations were applied where dichotomous predictors were involved to address hypotheses $H_{02}$–$H_{12}$ on associations between sociodemographic variables and KAP domains. For categorical variables with more than two groups (e.g., qualification, age group), one-way analysis of variance (ANOVA) was conducted when assumptions were met, while the Kruskal–Wallis test was applied otherwise. Where significant omnibus results were found, Tukey's HSD post hoc tests were performed to identify between-group differences.

Moreover, to examine hypotheses $H_{03}$–$H_{13}$ regarding the predictive value of sociodemographic factors on KAP scores, multiple linear regression analyses were conducted with KAP as dependent variables. Independent variables included age, gender, professional role, qualification, and years of experience. Model fit was evaluated using $R^2$, adjusted $R^2$, $F$ statistics, and associated $p$ values. Cohen's $f^2$ effect sizes were calculated to contextualize the strength of associations (small = 0.02, medium = 0.15, large = 0.35).

All analyses were two-tailed with significance set at $p < 0.05$. For correlational analyses, significance was additionally reported at both the 0.05 and 0.01 levels to highlight stronger associations. Effect sizes (Cohen's $d$, $\eta^2$, $\beta$ coefficients, odds ratios, and 95% confidence intervals, as appropriate) were reported to complement significance testing.

### Ethics statements

This study was approved by the Bioethics Committee of the Faculty of Biological Sciences at Quaid-i-Azam University, Islamabad (ref. #: BEC-FBS-QAU2021–270) and the ethical review committee of the Punjab Institute of Cardiology (PIC), Hospital in Lahore (ref. #: RTPGME-Research-204) in compliance with the Declaration of Helsinki and the Australian New Zealand registry. PIC granted a waiver of consent for the collection of administrative data for each randomized participants. Written informed consent was also obtained prior to enrollment in the study.

## Results

A total of 259 HCPs were invited to participate in the study. Of these, 117 were excluded: 51 did not meet the inclusion criteria, and 45 declined to participate. The remaining 82 participants completed the questionnaire. After excluding 6 incomplete responses, a total of 76 complete responses were included in the final analysis. The details of participant inclusion and exclusion are presented in the participant flow diagram (Fig 1).

In the final sample, 42 participants (55.3%) were cardiologists, while 34 participants (44.7%) were clinical pharmacists. Out of 76 study participants, 41 (53.9%) were female, 39 (51.9%) were aged between 28 and 33 years, 43 (56.6%) had attained post-graduate degrees or higher, 46 (60.5%) were employed in public hospitals, and 37 (44.7%) had 1–5 years of clinical experience. The socio-demographic data is presented in Table 1.

### HCPs Knowledge regarding GDMT

This study indicates that the accuracy percentages for the GDMT knowledge items varied from 22.7% to 80.25% (S1 Table). The average score of knowledge was 18.64 (SD: 2.02, range: 22–13), indicating an overall accurate rate of 84% (18.64/22*100) on this assessment (S2 Table). Moreover, a high proportion of participants (80.3%) reported being familiar with current clinical guidelines for managing IHD suggesting strong general awareness. Alarmingly, just 22.7% reported awareness of recent updates to GDMT in the past two years, highlighting a critical gap in current knowledge (S1 Table).

The Shapiro Wilk test indicated that knowledge scores were approximately normally distributed for both cardiologists ($p = 0.050$) and pharmacists ($p = 0.113$) (Table 2).

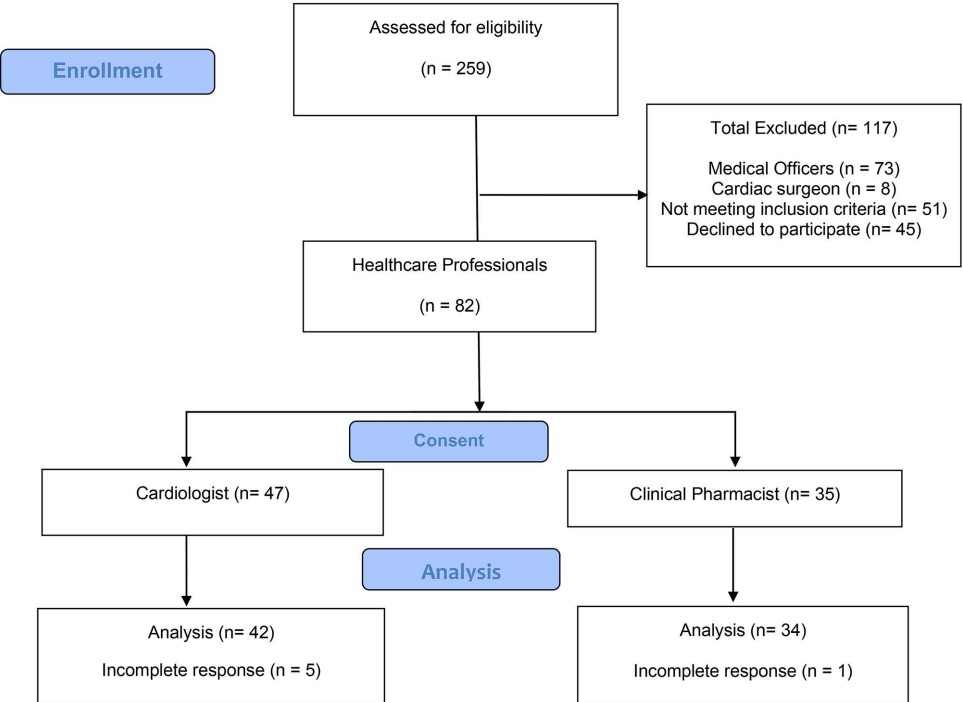

**Fig 1. Participant flow diagram on KAP toward GDMT.**

**Table 1. Socio-demographic data of the respondents.**

| Variables | Categories | N (%) | Mean ± Standard Deviation |
|---|---|---|---|
| Gender | Male | 35 (46.1) | 1.54 ± 0.502 |
| | Female | 41 (53.9) | |
| Age in year | 22–27 | 22 (28.9) | 1.91 ± 0.696 |
| | 28–33 | 39 (51.3) | |
| | 34–39 | 15 (19.7) | |
| Education | Graduation | 33 (43.4) | 1.57 ± 0.499 |
| | Post–graduation | 43 (56.6) | |
| Practice settings | Hospital | 46 (60.5) | 1.39 ± 0.492 |
| | Clinic | 30 (39.5) | |
| HCPs | Cardiologist | 42 (55.3) | 1.45 ± 0.501 |
| | Clinical Pharmacist | 34 (44.7) | |
| Experience | Less than 1 year | 22 (28.9) | 1.93 ± 0.718 |
| | 1–5 years | 37 (48.7) | |
| | 6–10 years | 17 (22.4) | |

Independent samples t-tests were performed to examine differences in GDMT-related knowledge scores across demographic groups. The results shows that cardiologists (M = 19.54, SD = 1.34) demonstrated significantly higher GDMT knowledge scores compared to clinical pharmacists (M = 17.52, SD = 2.17), $t(74) = 4.95$, $p < 0.001$, with unequal variances (*Levene's* F = 14.38, $p < 0.001$). These findings support the rejection of the null hypothesis ($H_{01}$) and acceptance of the alternative hypothesis ($H_{11}$).

**Table 2. Test of Normality of KAP Scores.**

| KAP Scores | HCPs | Shapiro-Wilk | | |
|---|---|---|---|---|
| | | Statistic | df | Sig. |
| Knowledge Score | Cardiologist | 0.947 | 42 | 0.050 |
| | Clinical Pharmacist | 0.949 | 34 | 0.113 |
| Attitude Score | Cardiologist | 0.909 | 42 | 0.003 |
| | Clinical Pharmacist | 0.936 | 34 | 0.048 |
| Practice Score | Cardiologist | 0.850 | 42 | 0.000 |
| | Clinical Pharmacist | 0.748 | 34 | 0.000 |

In addition, a statistically significant moderate negative correlation was observed between professional role and knowledge scores, $r = -0.499$, $p < .001$ (S3 Table), further confirming that cardiologists demonstrated higher knowledge compared with pharmacists (Fig 2).

However, participants with postgraduate education scored significantly higher (M = 19.09, SD = 1.84) than those with only a graduate degree (M = 18.06, SD = 2.12), $t(74) = -2.26$, $p = 0.027$, indicating that advanced education is associated with greater knowledge of GDMT. Accordingly, the null hypothesis ($H_{02}$) was rejected and the alternative hypothesis ($H_{12}$) was supported. Furthermore, a significant difference was also observed based on practice setting: HCPs working in hospital clinic setting (M = 19.56, SD = 1.50) had higher knowledge scores compared to those in hospital settings (M = 18.04, SD = 2.10), $t(74) = -3.428$, $p = 0.001$; again rejecting $H_{02}$ in favor of $H_1$. Moreover, there was no significant difference in knowledge scores between male (M = 18.80, SD = 1.60) and female participants (M = 18.44, SD = 2.32), $t(74) = 0.958$, $p = 0.341$, despite unequal variances (*Levene's* F = 6.339, $p = 0.014$). For gender, therefore, the null hypothesis ($H_{02}$) was retained and the alternative hypothesis ($H_{12}$) was not supported. A detail description is illustrated in Table 3.

A one-way ANOVA demonstrated a significant effect of age on GDMT-related knowledge scores, $F(2, 73) = 8.13$, $p = .001$. Post hoc analysis using the Tukey HSD test indicated that participants aged 34–39 years had significantly higher knowledge scores compared to those aged 22–27 years (mean difference = 2.30, $p = .001$). These findings indicate that GDMT-related knowledge increases with age, particularly between the youngest and oldest groups. Accordingly, the null hypothesis ($H_{02}$) was rejected, and the alternative hypothesis ($H_{12}$) was supported.

In contrast, no significant differences were observed in knowledge scores across professional experience categories, $F(2, 73) = 1.17$, $p = .316$. However, tukey post hoc comparisons confirmed that none of the pairwise differences between experience groups reached statistical significance (all $p > .05$). These findings indicate that professional experience was

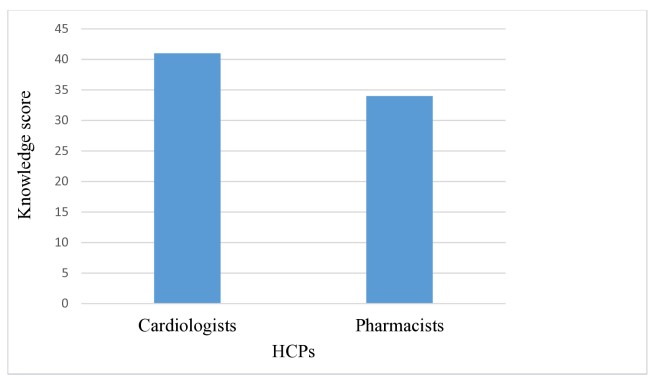

**Fig 2. Correlation of knowledge score regarding GDMT implementation between cardiologists and clinical pharmacists.**

**Table 3. Comparison of GDMT-related knowledge scores across demographic characteristics using independent sample t-test.**

| Variables | Categories | N (%) | Mean ± Standard Deviation | Levene's test for equality of variance | | t-test for equality of means | | |
|---|---|---|---|---|---|---|---|---|
| | | | | F | P value | t | df | P value |
| Gender | Male | 35 (46.1) | 18.8 ± 1.60 | 6.339 | 0.014 | 0.958 | 74 | 0.341 |
| | Female | 41 (53.9) | 18.4 ± 2.32 | | | | | |
| Education | Graduation | 33 (43.4) | 18.1 ± 2.12 | 0.458 | 0.501 | −2.26 | 74 | 0.027 |
| | Post−graduation | 43 (56.6) | 19.1 ± 1.84 | | | | | |
| Practice settings | Hospital | 46 (60.5) | 18.0 ± 2.10 | 4.768 | 0.032 | −3.428 | 74 | 0.001 |
| | Clinic | 30 (39.5) | 19.6 ± 1.50 | | | | | |
| HCPs | Cardiologist | 42(55.3) | 19.5 ± 1.34 | 14.38 | 0.000 | 4.95 | 74 | 0.00 |
| | Clinical Pharmacist | 34 (44.7) | 17.5 ± 2.17 | | | | | |
| Total Knowledge score | | 76 (100) | 18.6 ± 2.02 | | | | | |

not associated with GDMT-related knowledge, and therefore the null hypothesis (H$_{02}$) was retained while the alternative hypothesis (H$_{12}$) was not supported. A detail description of ANOVA results is summarized in Table 4.

### HCPs attitude toward GDMT

The present results show that majority of respondents agreed that implementing GDMT in ordinary clinical practice is possible, and that pharmacist's engagement enhances adherence to GDMT among IHD patients, with scores of 91.5% and 78.2%, respectively (S1 Table). However, limited consultation time (47.4%) and lack of patient adherence (25%) were the most reported barriers. The average attitude score is 10.42 (SD: 2.06, range: 14−7), indicating an overall accurate rate of 74% (10.42/14*100) on this assessment (S2 Table). The Shapiro Wilk test indicated that attitude scores showed deviation from normality in cardiologists ($p = 0.003$) and borderline deviation in pharmacists ($p = 0.048$) (Table 2).

The Mann-Whitney analysis indicated a significant difference was also observed between professional roles: cardiologists (Mean Rank = 43.25) reported more positive attitudes than pharmacists (Mean Rank = 32.63), U = 514.5, Z = −2.107,

**Table 4. Comparison of GDMT-related knowledge scores across demographics using one-way ANOVA and Tukey post hoc tests.**

| Variables | Tukey HSD | | | | | ANOVA | |
|---|---|---|---|---|---|---|---|
| | (I) Age categories | (J) Age categories | Mean Difference (I-J) | Std. Error | P value | F | P value |
| Age | 22–27 | 28–33 | −1.610* | 0.494 | 0.005 | 8.13 | 0.001** |
| | | 34–39 | −2.303* | 0.621 | 0.001 | | |
| | 28–33 | 22–27 | 1.610* | 0.494 | 0.005 | | |
| | | 34–39 | −0.692 | 0.563 | 0.441 | | |
| | 34–39 | 22–27 | 2.303* | 0.621 | 0.001 | | |
| | | 28–33 | 0.692 | 0.563 | 0.441 | | |
| Experience | Less than 1 year | 1–5 years | −0.800 | 0.543 | 0.310 | 1.170 | 0.316 |
| | | 6–10 years | −0.732 | 0.652 | 0.503 | | |
| | 1–5 years | Less than 1 year | 0.800 | 0.543 | 0.310 | | |
| | | 6–10 years | 0.068 | 0.591 | 0.993 | | |
| | 6–10 years | Less than 1 year | 0.732 | 0.652 | 0.503 | | |
| | | 1–5 years | −0.068 | 0.591 | 0.993 | | |

a. Dependent variables are age and experience

b. Std. error = Standard error

c. $p < 0.05$ = significant (*); $p < 0.01$ = highly significant (**).

$p = 0.035$. These findings support rejection of the null hypothesis ($H_{01}$) and acceptance of the alternative hypothesis ($H_{11}$). Similarly, a weak but significant negative correlation was observed between HCPs and attitude scores ($r = -0.262$, $p = 0.022$) (S3 Table), suggesting that cardiologists held more favorable attitudes toward GDMT implementation (Fig 3). These results provide evidence of an association between sociodemographic factors and attitudes, leading to rejection of the null hypothesis ($H_{02}$) and support for the alternative hypothesis ($H_{12}$).

A Mann–Whitney U test showed a statistically significant difference in attitude scores by educational level, with post-graduate professionals (Mean Rank = 44.17) reporting more favorable attitudes toward GDMT than graduates (Mean Rank = 31.11), $U = 465.5$, $Z = -2.59$, $p = .010$ (Table 5). This result supports rejection of the null hypothesis ($H_{02}$) and acceptance of the alternative hypothesis ($H_{12}$) for education. In contrast, no significant differences in attitude scores were observed by gender ($U = 579.0$, $p = .144$) or practice setting ($U = 627.5$, $p = .502$) (Table 5), indicating that these variables were not associated with differing attitudes toward GDMT use in IHD management. For these comparisons, the null hypothesis ($H_{02}$) was retained.

The Kruskal–Wallis H test analysis revealed a statistically significant difference in attitude scores across age groups, $\chi^2(2) = 19.29$, $p < 0.001$. Participants aged 28–33 years had the highest mean rank (49.22), while those aged 22–27 and 34–39 years had substantially lower mean ranks (27.39 and 26.93, respectively), suggesting that professionals in the 28–33 age group exhibited more favorable attitudes toward GDMT implementation. Accordingly, the null hypothesis ($H_{02}$) was rejected, and the alternative hypothesis ($H_{12}$) was supported.

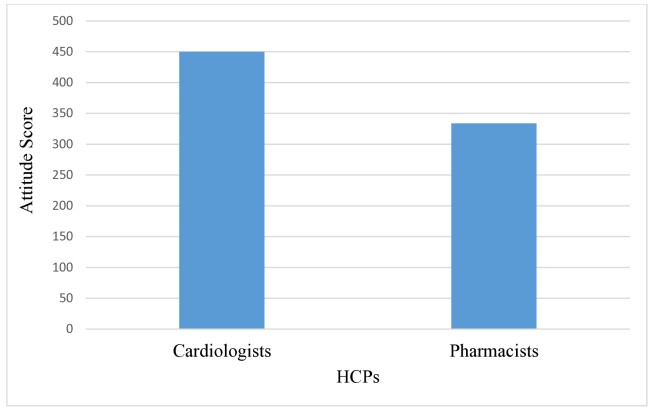

**Fig 3. Correlation of attitude score regarding GDMT implementation between cardiologists and clinical pharmacists.**

**Table 5. Comparison of GDMT-related attitude scores across demographics using Mann-Whitney test.**

| Variables | Categories | N | Mean Rank | U | Z | P value |
|---|---|---|---|---|---|---|
| Gender | Male | 35 | 42.46 | 579.00 | −1.46 | 0.144 |
| | Female | 41 | 35.12 | | | |
| Education | Graduation | 33 | 31.11 | 465.5 | −2.58 | 0.010* |
| | Post-graduation | 43 | 44.17 | | | |
| Practice settings | Hospital | 46 | 37.14 | 627.5 | −0.67 | 0.502 |
| | Clinic | 30 | 40.58 | | | |
| HCPs | Cardiologist | 42 | 43.25 | 514.500 | −2.10 | 0.035 |
| | Clinical Pharmacist | 34 | 32.63 | | | |

a. U = Mann–Whitney

b. $p < 0.05$ = significant (*); $p < 0.01$ = highly significant (**).

Similarly, attitude scores differed significantly by professional experience, $\chi^2(2) = 7.219$, $p = 0.027$ (Table 6). Participants with 1–5 years of experience had higher mean ranks (42.34) compared to those with 6–10 years of experience (25.97), indicating that early-career professionals reported more favorable attitudes than their more experienced counterparts (Table 6). These findings indicate that attitudes toward GDMT were influenced by both age and professional experience, with younger and less experienced HCPs demonstrating more favorable attitudes toward its implementation. Accordingly, the null hypothesis ($H_{02}$) was rejected, and the alternative hypothesis ($H_{12}$) was supported.

Forty-five percent of study participants reevaluated a patient's GDMT solely in response to changes in clinical condition, whereas 56.5% examined patient adherence to GDMT through direct inquiry during follow-up (S1 Table). The average practice score is 9.51 (SD: 2.55, range: 14–7), indicating an overall accurate rate of 68% (9.51/14*100) on this assessment (S2 Table). The Shapiro Wilk test indicated that Practice scores were significantly non-normal in both groups ($p < 0.001$) (Table 2).

However, the Mann–Whitney U test indicated no statistically significant difference in practice scores between cardiologists and pharmacists, $U = 645.0$, $Z = -0.74$, $p = .459$ (Table 7), suggesting comparable engagement with GDMT-related practices across both groups. Accordingly, the null hypothesis ($H_{01}$) was retained. The point-biserial correlations results also confirmed that there is no significant correlation was observed between HCPs and practice scores ($r = -0.088$, $p = 0.449$) (S3 Table), indicating no meaningful difference in self-reported practice behavior (Fig 4).

Moreover, practice scores did not differ significantly between male and female participants, $U = 656.5$, $Z = -0.653$, $p = 0.514$, and hospital-based and clinic-based practitioners, $U = 642.0$, $Z = -0.524$, $p = 0.600$ (Table 7). Overall, these findings suggest that GDMT-related practice behaviors were consistent across gender, educational status, practice setting, and healthcare profession, with no group demonstrating significantly greater involvement. Accordingly, the null hypotheses ($H_{01}$ and $H_{02}$) were retained, indicating no meaningful variation in practice across these demographic factors.

However, the Kruskal-Wallis H indicated that age groups showed a statistically significant difference in practice scores, $\chi^2(2) = 11.348$, $p = 0.003$. Participants aged 28–33 years had the highest mean rank (46.51), followed by those aged 34–39 years (32.00) and 22–27 years (28.73). This indicates that mid-career professionals may exhibit more favorable practice toward GDMT. Accordingly, the null hypothesis ($H_{02}$) was rejected, and the alternative hypothesis ($H_{12}$) was supported. In contrast, professional experience was not significantly associated with differences in practice scores, $\chi^2(2) = 0.065$, $p = 0.968$. Mean ranks were comparable across experience groups (1–5 years = 38.43; 6–10 years = 39.56; 22–27 years = 37.80) (Table 8), suggesting that years of experience did not meaningfully influence practice toward GDMT in this sample. Therefore, the null hypothesis ($H_{02}$) was retained, and the alternative hypothesis ($H_{12}$) was not supported.

**Table 6. Comparison of GDMT-related attitude scores across demographics using Kruskal Wallis Test.**

| Variables | Categories | N | Mean Rank | Chi-square | df | P value |
|---|---|---|---|---|---|---|
| Age in year | 22–27 | 22 | 27.39 | 19.29 | 2 | 0.000** |
| | 28–33 | 39 | 49.22 | | | |
| | 34–39 | 15 | 26.93 | | | |
| Experience in year | 22–27 | 22 | 27.39 | 7.219 | 2 | 0.027 |
| | 1–5 years | 37 | 42.34 | | | |
| | 6–10 years | 17 | 25.97 | | | |
| | Total | 76 | | | | |

a. Kruskal Wallis Test

b. Grouping Variable: Age and Experience

c. Dependent variables are attitude

d. $p < 0.05$ = significant (*); $p < 0.01$ = highly significant (**).

**Table 7. Comparison of GDMT-related practice scores across demographics using Mann-Whitney test.**

| | | N | Mean Rank | U | Z | P value |
|---|---|---|---|---|---|---|
| Gender | Male | 35 | 36.76 | 656.5 | −0.653 | 0.514 |
| | Female | 41 | 39.99 | | | |
| Educational level | Graduation | 33 | 34.17 | 566.5 | −1.540 | 0.124 |
| | Post-graduation | 43 | 41.83 | | | |
| Practice settings | Hospital | 46 | 37.46 | 642.0 | −0.524 | 0.600 |
| | Clinic | 30 | 40.10 | | | |
| HCPs | Cardiologist | 42 | 40.14 | 645.0 | −0.741 | 0.459 |
| | Clinical Pharmacist | 34 | 36.47 | | | |

a. U = Mann–Whitney

b. *p* < 0.05 = significant (*); *p* < 0.01 = highly significant (**).

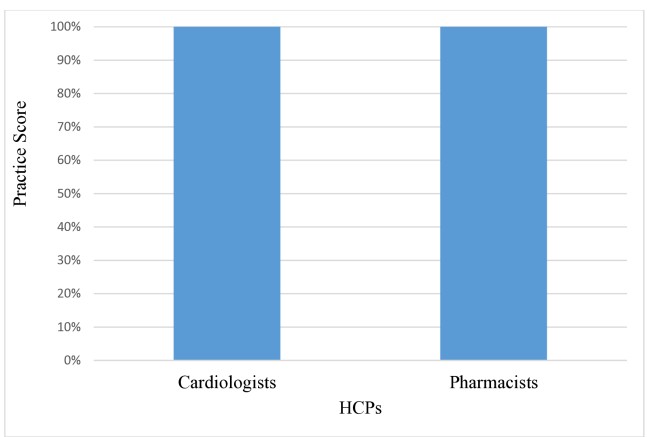

**Fig 4. Correlation of practice score regarding GDMT implementation between cardiologists and clinical pharmacists.**

Multiple linear regression analyses were conducted to evaluate the influence of demographic and professional characteristics on KAP scores related to GDMT. The model predicting knowledge scores was statistically significant, $F(6, 69) = 9.90$, $p < .001$, accounting for 46.3% of the variance ($R^2 = .463$, Adjusted $R^2 = .416$). The model predicting attitude scores was also significant, $F(6, 69) = 4.49$, $p = .001$, explaining 28.1% of the variance ($R^2 = .281$, Adjusted $R^2 = .218$). By contrast, the regression model predicting practice scores was not significant, $F(6, 69) = 0.27$, $p = .948$, with minimal explained variance ($R^2 = .023$, Adjusted $R^2 = −.062$), indicating that the included predictors did not account for meaningful variability in practice outcomes (S4 Table). These findings support the rejection of the null hypothesis ($H_{03}$) and acceptance of the alternative hypothesis ($H_{13}$) for knowledge and attitude, while the null hypothesis was retained for practice, suggesting that demographic and professional characteristics significantly predicted knowledge and attitudes but not practice toward GDMT.

## Knowledge

Within the multiple linear regression model, age demonstrated a strong positive association with knowledge ($\beta = 0.528$, $p < .001$), with a large effect size ($f^2 = 0.38$), indicating a substantial contribution to the explained variance. Professional role was also a significant predictor, with a negative association ($\beta = −0.508$, $p < .001$) and a large effect size ($f^2 = 0.371$), suggesting that differences between cardiologists and clinical pharmacists accounted for a meaningful proportion of variance

**Table 8. Comparison of GDMT-related attitude scores across demographics using Kruskal Wallis Test.**

| Variables | Categories | N | Mean Rank | Chi-square | df | P value |
|---|---|---|---|---|---|---|
| Age in year | 22–27 | 22 | 28.73 | 11.348 | 2 | 0.003* |
|  | 28–33 | 39 | 46.51 |  |  |  |
|  | 34–39 | 15 | 32.00 |  |  |  |
|  | 22–27 | 22 | 37.80 | 0.065 | 2 | 0.968 |
|  | 1–5 years | 37 | 38.43 |  |  |  |
|  | 6–10 years | 17 | 39.56 |  |  |  |
|  | Total | 76 |  |  |  |  |

a. Kruskal Wallis Test

b. Grouping Variable: Age and Experience

c. Dependent variables are attitude

d. $p < 0.05$ = significant (*); $p < 0.01$ = highly significant (**).

in knowledge scores. In contrast, gender, education, practice setting, and years of experience were not statistically significant predictors, each demonstrating small or negligible effect sizes ($f^2 \leq 0.081$). These results provide partial support for the alternative hypothesis ($H_{13}$), indicating that certain demographic and professional characteristics significantly predicted knowledge, whereas others did not.

## Attitude

With respect to attitude, multiple linear regression model shows that education ($\beta = 0.285$, $p = 0.018$), gender ($\beta = -0.213$, $p = 0.044$), HCPs ($\beta = -0.232$, $p = 0.048$), and experience ($\beta = -0.396$, $p = 0.002$) were statistically significant predictors of attitude toward GDMT. Among these, experience had the largest effect size ($f^2 = 0.130$), approaching a medium effect, while the others contributed small effects ($f^2$ ranging from 0.043 to 0.069). Age and practice setting were non-significant and had trivial or negative effect sizes (e.g., $f^2 = -0.773$ and $-0.012$, respectively), indicating no meaningful contribution to variance in attitude scores. These findings partially support the alternative hypothesis ($H_{13}$), demonstrating that certain demographic and professional characteristics significantly predicted attitudes toward GDMT, whereas others showed no meaningful influence.

## Practice

Professional role ($\beta = -0.232$, $p = .048$), qualification ($\beta = 0.285$, $p = .018$), gender ($\beta = -0.213$, $p = .044$), and experience ($\beta = -0.396$, $p = .002$) emerged as significant predictors of self-reported GDMT-related practice behaviors. Among these, professional role demonstrated a large effect size ($f^2 = 0.419$), indicating a substantial contribution to differences in practice scores. By contrast, the effect sizes for qualification, gender, and experience were small (all $f^2 < 0.08$), suggesting limited practical impact despite their statistical significance (Table 9). These results therefore provide partial support for the alternative hypothesis ($H_{13}$), highlighting that while certain predictors significantly influenced practice, their overall contribution to variance was modest.

Among the predictors analyzed, age and HCPs were significant determinants of knowledge. Older participants demonstrated higher knowledge scores, indicating that greater clinical exposure and cumulative experience contribute to stronger familiarity with GDMT principles. Cardiologists and clinical pharmacists showed higher knowledge levels than other healthcare cadres, reflecting their direct involvement in cardiovascular patient care.

In the attitude model, education level, gender, and experience were significant predictors. Participants with postgraduate education reported more favorable attitudes toward GDMT, suggesting that advanced academic training fosters

**Table 9. Association of sociodemographic variables with GDMT of KAP Scores by multiple linear regression.**

| Model | Unstandardized Coefficients | | Standardized Coefficients | T | P value | 95.0% Confidence Interval for B | | Cohen's $f^2$ |
|---|---|---|---|---|---|---|---|---|
| | B | Std. Error | Beta | | | Lower Bound | Upper Bound | |
| **Knowledge** | | | | | | | | |
| Age | 1.537 | .0344 | 0.528 | 4.468 | 0.000** | 0.851 | 2.223 | 0.38 |
| Gender | −0.285 | 0.362 | −0.071 | −0.788 | 0.433 | −1.008 | 0.437 | 0.081 |
| Education | −0.129 | 0.413 | −0.032 | −0.311 | 0.757 | −0.953 | 0.696 | 0.074 |
| Practice setting | 0.187 | 0.440 | 0.045 | 0.424 | 0.673 | −0.691 | 1.065 | 0.074 |
| HCPs | −2.054 | 0.404 | −0.508 | −5.090 | 0.000** | −2.860 | −1.249 | 0.371 |
| Experience | −0.503 | 0.303 | −0.178 | −1.658 | 0.102 | −1.107 | 0.102 | 0.014 |
| **Attitude** | | | | | | | | |
| Age | 0.552 | 0.406 | 0.186 | 1.359 | 0.178 | −0.258 | 1.363 | −0.773 |
| Gender | −0.876 | 0.428 | −0.213 | −2.048 | 0.044 | −1.729 | −0.023 | 0.044 |
| Education | 1.180 | 0.488 | 0.285 | 2.416 | 0.018 | 0.206 | 2.154 | 0.069 |
| Practice setting | −0.201 | 0.520 | −0.048 | −0.387 | 0.700 | −1.238 | 0.836 | −0.012 |
| HCPs | −0.960 | 0.477 | −0.232 | −2.013 | 0.048 | −1.911 | −0.009 | 0.043 |
| Experience | −1.140 | 0.358 | −0.396 | −3.185 | 0.002* | −1.854 | −0.426 | 0.130 |
| **Practice** | | | | | | | | |
| Age | 0.552 | 0.406 | 0.186 | 1.359 | 0.178 | −0.258 | 1.363 | 0.068 |
| Gender | −0.876 | 0.428 | −0.213 | −2.048 | 0.044 | −1.729 | −0.023 | 0.068 |
| Education | 1.180 | 0.488 | 0.285 | 2.416 | 0.018 | 0.206 | 2.154 | 0.076 |
| Practice setting | −0.201 | 0.520 | −0.048 | −0.387 | 0.700 | −1.238 | 0.836 | 0.071 |
| HCPs | −0.960 | 0.477 | −0.232 | −2.013 | 0.048 | −1.911 | −0.009 | 0.419 |
| Experience | −1.140 | 0.358 | −0.396 | −3.185 | 0.002* | −1.854 | −0.426 | 0.067 |

$p < 0.05$ = significant (*); $p < 0.01$ = highly significant (**).

evidence-based decision-making. Female respondents demonstrated slightly lower attitude scores, which might relate to fewer opportunities for clinical specialization in cardiology practice settings. Moreover, longer professional experience was associated with less favorable attitudes, possibly reflecting attitudinal inertia or lower engagement in recent continuing education programs.

For the practice domain, education, gender, and professional category remained significant. Higher educational attainment predicted better GDMT implementation in daily practice, while variations across professions again highlighted the pivotal role of pharmacists and physicians in bridging knowledge with clinical action.

## Discussion

The management of IHD is a complex process that requires a multidisciplinary approach to optimize patient outcomes [6]. Clinical pharmacists play a crucial role alongside cardiologists as a healthcare team in implementing GDMT for individuals with IHD [33]. The partnership between clinical pharmacists and cardiologists is the effective for implementation of GDMT in patients with IHD [34]. To best of our knowledge, there is a lack of available studies examining the KAP of HCPs regarding the implementation of GDMT particularly among cardiologists and clinical pharmacists [35,36].

This KAP survey revealed that cardiologists in Lahore, Pakistan, demonstrated higher knowledge and more favorable attitudes toward guideline-directed medical therapy (GDMT) for IHD than clinical pharmacists, although both groups reported comparable levels of guideline-based practice. These findings reflect global trends showing strong clinician awareness of GDMT principles but suboptimal translation into practice [35,37]. The association between higher

educational attainment and greater knowledge aligns with reports from China and Malaysia, where postgraduate training and specialization predicted superior GDMT familiarity [37,38]. In contrast, knowledge gaps among pharmacists observed in our cohort mirror findings from Jordan and Saudi Arabia, where pharmacists displayed limited understanding of cardiovascular guideline content [39,40].

Age and experience were also significant predictors of knowledge. Older and more experienced clinicians achieved higher scores, likely reflecting cumulative clinical exposure and continuing medical education. Similar associations have been reported in Ethiopian and Middle Eastern studies, which attributed improved GDMT familiarity to professional longevity and exposure to specialized training environments [39,41]. However, younger practitioners in our sample showed strong baseline knowledge and positive attitudes, suggesting that recent curricula emphasizing evidence-based practice have improved early-career competence. Comparable generational differences have been noted among younger clinicians in Lithuania, Bangladesh, and China, where new graduates exhibit greater openness to evidence-based and team-based care models [42].

Despite strong knowledge and attitudes, overall self-reported practice remained moderate (~68%), underscoring the persistent "know–do" gap documented globally [43]. Several system-level and behavioral factors likely contribute. Nearly half of respondents cited limited consultation time and poor patient adherence as barriers to GDMT implementation concerns echoed internationally. In a recent ESC survey, physicians identified patient adherence, polypharmacy, and side-effect concerns as leading obstacles to optimal GDMT use [44]. A meta-analysis confirmed that clinician workload, patient non-adherence, and fragmented care systems remain universal impediments to guideline-driven therapy [35].

Pharmacist involvement in multidisciplinary care is increasingly recognized as a key driver of GDMT adherence [34,45]. Evidence from high-income settings shows that pharmacist-led or pharmacist-integrated heart-failure clinics significantly improve medication optimization and dose titration [40,46]. However, similar initiatives in LMICs are rare. Our finding that pharmacists' GDMT practice scores did not differ significantly from cardiologists' may reflect limited institutional frameworks enabling pharmacists' active clinical participation. Establishing pharmacist-led cardiovascular clinics or structured interprofessional training programs could enhance adherence and optimize patient outcomes [47].

Emerging digital tools may also support real-time GDMT monitoring and education. Digital decision-support systems and audit-feedback interventions have improved adherence to HF guidelines in multicenter trials [48–50]. Incorporating such technologies into LMIC healthcare systems could help overcome time constraints and ensure standardized, up-to-date therapy adjustments.

Overall, this study provides evidence of strong GDMT awareness among Pakistani cardiologists and pharmacists but identifies structural and educational barriers limiting full implementation. Addressing these gaps will require system-level interventions, including structured continuing professional development, pharmacist-led clinics, and integration of digital adherence tools. Aligning such strategies with local health-system capacity may help translate knowledge into practice and narrow GDMT disparities between LMIC and high-income settings.

### Implications for clinical practice

These insights have important implications. First, the profession-based knowledge gap suggests strengthening training and continuing education for pharmacists and early-career HCPs. Hands-on workshops or modules on GDMT could help, as recommended by training-focused studies [1,2]. Second, the near-universal belief in guideline relevance and pharmacists' role is encouraging; it implies that multidisciplinary team approaches (heart-failure clinics, pharmacist-led stewardship programs, etc.) could be well received. Indeed, integrated models have shown promise a pharmacist-assisted heart failure clinic was able to prescribe GDMT at rates comparable to a standard clinic [7]. Third, addressing barriers like limited time and patient adherence will require system-level strategies, longer consultations for complex cases, follow-up support for medication adherence, and locally adapted guidelines that acknowledge resource constraints (as advocated by WHO roadmaps and recent surveys [4,9]. Furthermore, the integration of pharmacist-led clinics, telemonitoring

platforms, and structured patient education programs may enhance continuity of GDMT delivery to mitigate implementation barriers such as limited consultation time and poor follow-up adherence, particularly in resource-constrained settings. Finally, future research and interventions should be conducted with rigor and transparency to ensure findings are reproducible and generalizable.

## Recommendations for practical implications

Building on the practical implications of this study, several specific recommendations can be made to strengthen GDMT implementation in resource-limited settings:

**Targeted Professional Development.** Training should move beyond general awareness sessions and incorporate interactive, case-based modules, simulation-based learning, and periodic refresher courses. Tailoring content to the specific needs of cardiologists and pharmacists could help address role-specific gaps and improve inter-professional understanding of GDMT.

**Institutional Support for Implementation.** Hospitals should embed guideline adherence into quality assurance frameworks. This may include electronic prescribing systems with GDMT prompts, regular audit-and-feedback cycles, and inclusion of GDMT indicators in hospital performance metrics.

**Enhancing Inter-Professional Collaboration.** Multidisciplinary teams involving physicians, pharmacists, nurses, and allied health professionals should be formally integrated into IHD management. Regular joint case discussions and ward rounds could foster shared responsibility, reduce attitudinal barriers, and promote consistent guideline adoption.

**Policy and System-Level Interventions.** Health authorities should ensure sustainable access to essential cardiovascular medicines by strengthening supply chain mechanisms and adopting pooled procurement strategies. Additionally, incorporation of GDMT adherence indicators into national cardiovascular disease programs would facilitate monitoring and accountability.

**Research and Evaluation.** Future studies should employ larger, multi-center designs to improve generalizability and external validity. Longitudinal and interventional research would be particularly valuable to evaluate the impact of targeted educational and institutional interventions on clinical outcomes.

## Strength and limitation of study

**Strengths.** This study is among the first in Pakistan to compare cardiologists and pharmacists on GDMT-related KAP, filling a local data gap. The survey covered multiple GDMT dimensions and used appropriate statistical tests (Kruskal–Wallis, regression) to adjust for confounders. A relatively high response rate (survey completion of 80–90%) enhances confidence in the findings. The mixed professional sample (cardiologists vs. pharmacists) and inclusion of demographic variables offer insights into factors influencing GDMT knowledge and attitudes.

**Limitations.** This study also presents certain limitations. This study was limited by its single-center design and reliance on a small, convenience-based sample (n = 76) which may reduce the generalizability of findings and introduce selection bias. However, a larger, multi-center study would strengthen external validity. Recruitment was dependent on the availability and willingness of eligible cardiologists and clinical pharmacists during the study period. As such, individuals who were more accessible or engaged with clinical education activities may have been more likely to participate, introducing potential selection bias. Furthermore, this approach restricted the final sample size, as participation was not randomized and was influenced by operational constraints within the hospital setting, such as clinical workload and time availability. These factors may have affected the representativeness of the sample and the distribution of certain demographic characteristics. Furthermore, the absence of an independent interviewer may also be considered a limitation. The interviewer was familiar with the majority of participants prior to this study, and all participants were aware at the time of interview that

she was a pharmacist with the objective of developing medicines review services. This may have influenced participants' responses to certain questions. The participant pool was restricted to a tertiary-care institution, and thus may not represent HCPs working in non-specialized or rural settings. Future multi-center studies with larger, randomly selected samples are recommended to validate these findings and enhance external validity. Furthermore, the use of self-administered questionnaires introduces potential response and social desirability bias, whereby participants may have overstated their knowledge or adherence to GDMT practices to align with perceived professional norms.

## Conclusion

This KAP survey found that cardiologists in Lahore, Pakistan, demonstrated higher knowledge and more favorable attitudes toward GDMT for IHD than clinical pharmacists, although both groups reported similar levels of guideline-based practice. The findings highlight persistent knowledge-practice gaps and emphasize the need for structured GDMT education targeting pharmacists and early-career clinicians. To enhance adherence, health systems should implement multidisciplinary training and pharmacist-led cardiovascular care models that support collaborative decision-making and patient counselling. Integrating continuing professional development, digital decision-support tools, and feedback mechanisms could strengthen guideline implementation and consistency in care. In LMICs, empowering pharmacists through enhanced clinical roles and system-level support can help translate GDMT awareness into sustainable practice improvements, ultimately improving cardiovascular outcomes and health system performance.

### Recommendation for future studies

While this survey captures broad patterns in GDMT-related behaviors, it does not explore the underlying reasons for suboptimal adoption. Future qualitative research may help elucidate context-specific barriers such as institutional inertia, therapeutic uncertainty, or perceived lack of patient receptivity factors reported in other LMIC contexts. However, improved adherence to GDMT through augmented interdisciplinary cooperation could play a crucial role in optimizing care for IHD patients. The subsequent multi-centered research should investigate the long-term impact of such interventions on patient outcomes and evaluate the efficacy of structured training programs in mitigating KAP gaps in clinical practice. Future studies should adopt larger, multi-center designs across diverse healthcare settings to enhance the external validity and generalizability of findings on GDMT implementation. The subsequent investigation will elucidate strategies to enhance multidisciplinary collaboration and improve adherence to evidence-based treatment guidelines, ultimately contributing to improved clinical outcomes for patients with IHD.

## Supporting information

**S1 File. Knowledge, attitudes, and practices toward guideline-directed medical therapy in ischemic heart disease (KAP-GDMT-IHD) questionnaire.**
(PDF)

**S2 File. STROBE checklist for cross sectional study design.**
(DOCX)

**S1 Table. Summary of questions for knowledge, attitudes and practices towards GDMT.** The correct answer is presented in percentage.
(DOCX)

**S2 Table. Descriptive statistics of KAP.**
(DOCX)

**S3 Table. Correlation of HCPs vs KAP scores.**
(DOCX)

**S4 Table. Multiple linear regression analysis predicting KAP toward GDMT among cardiologists and pharmacists.**
(DOCX)

## Author contributions

**Conceptualization:** Umm-e- Kalsoom, Amjad Khan, Syed Sikandar Shah, Ayesha Iqbal, Tahir Mehmood, Syed Mansoor Ahmed, Noshed Khan, Yu Fang.

**Data curation:** Umm-e- Kalsoom, Ayesha Iqbal, Tahir Mehmood, Syed Mansoor Ahmed.

**Formal analysis:** Umm-e- Kalsoom, Tahir Mehmood.

**Investigation:** Umm-e- Kalsoom, Tahir Mehmood, Syed Mansoor Ahmed, Noshed Khan.

**Methodology:** Umm-e- Kalsoom, Ayesha Iqbal, Tahir Mehmood, Syed Mansoor Ahmed.

**Project administration:** Umm-e- Kalsoom.

**Resources:** Umm-e- Kalsoom.

**Software:** Umm-e- Kalsoom.

**Supervision:** Amjad Khan, Yu Fang.

**Validation:** Umm-e- Kalsoom, Amjad Khan, Syed Sikandar Shah, Ayesha Iqbal, Noshed Khan, Yu Fang.

**Visualization:** Umm-e- Kalsoom, Amjad Khan, Syed Sikandar Shah, Ayesha Iqbal, Noshed Khan.

**Writing – original draft:** Umm-e- Kalsoom, Syed Sikandar Shah.

**Writing – review & editing:** Umm-e- Kalsoom, Amjad Khan, Syed Sikandar Shah, Ayesha Iqbal, Tahir Mehmood, Noshed Khan, Yu Fang.

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
