## [Decision Letter · Decision Letter 0]

24 Jun 2025

PONE-D-25-18404Implementation of Guideline-Directed Medical Treatment for Ischemic Heart Disease Management: A Knowledge, Attitude and Practice Based Cross-Sectional SurveyPLOS ONE

Dear Dr. Khan,

Thank you for submitting your manuscript to PLOS ONE. After careful consideration, we feel that it has merit but does not fully meet PLOS ONE’s publication criteria as it currently stands. Therefore, we invite you to submit a revised version of the manuscript that addresses the points raised during the review process.

We look forward to receiving your revised manuscript.

Kind regards,

Mohammed Abutaleb, PhD

Academic Editor

PLOS ONE

Journal Requirements:

[No authors have competing interest].

Reviewers' comments:

Reviewer's Responses to Questions

**Comments to the Author**

1. Is the manuscript technically sound, and do the data support the conclusions?

Reviewer #1: No

Reviewer #2: Partly

2. Has the statistical analysis been performed appropriately and rigorously?

Reviewer #1: No

Reviewer #2: No

3. Have the authors made all data underlying the findings in their manuscript fully available?

Reviewer #1: No

Reviewer #2: Yes

4. Is the manuscript presented in an intelligible fashion and written in standard English?

Reviewer #1: No

Reviewer #2: Yes

5. Review Comments to the Author

Reviewer #1: This study addresses an important area regarding the implementation of Guideline-Directed Medical Therapy (GDMT) implementation for ischemic heart disease (IHD) within resource-limited settings. The finding that "this is the first investigation of the knowledge, attitudes, and practices of HCPs regarding the implementation of GDMT in patients with IHD" is a strong point. However, the manuscript needs to be strengthened in terms of methodological rigor, statistical reporting, and clarity of presentation. Limitations include a single-center design (Punjab Institute of Cardiology) and small convenience sample (n=76), restrict generalizability and risk selection bias. Pharmacist (n=34) versus cardiologist (n=42) comparisons may lack statistical power, while inconsistent practice metrics (e.g., citing highest scores for participants over 50 years despite Table 4 excluding this age group) and absent correlation figures weaken the evidence for healthcare professional (HCP) differences. Broader claims about collaborative care necessitate multi-center validation.

Statistical methods (t-tests, ANOVA, regression) are generally appropriate for the cross-sectional design, and Bonferroni correction (α=0.0125) is correctly applied. However, critical anomalies undermine rigor: Table 3 reports implausible attitude scores (e.g., 34.395 ± 0.954 for post-graduates), suggesting data errors. Power analysis results are inconsistently interpreted (e.g., 0.949 for practice vs. 0.617 for knowledge) without justification for variable thresholds, and β-values (e.g., β=0.383 for knowledge disparity) lack effect-size context (e.g., Cohen’s), obscuring clinical significance. Additionally, while the manuscript claims all data are included, only supplementary materials (questionnaires, STROBE checklist) are accessible, and raw datasets are unavailable. Authors should rectify statistical tables and share de-identified data via repositories.

The manuscript is generally clear but contains grammatical errors (e.g., "re you confident" corrected to "Are you confident"; inconsistent "perception" vs. "practice" scoring) and ambiguous phrasing (e.g., contradictory β-values describing attitude differences). Terminology like "bad KAP" should be standardized to "poor KAP." Ethically, approval and consent procedures are documented, though interviewer-participant familiarity risks social desirability bias—acknowledged by authors. As the first Pakistani KAP study on GDMT, it valuably highlights barriers (e.g, time constraints) and aligns with global calls for multidisciplinary collaboration. However, small sample size limits subgroup analyses, and qualitative insights into poor GDMT adoption are absent. Revisions should clarify statistical anomalies, propose barrier-mitigation strategies (e.g., telehealth), and contextualize findings via LMIC comparisons.

Reviewer #2: The manuscript requires major revisions, particularly, the statistical analysis and interpretation. The research question was not stated explicitly nor were subquestions identified. A hypothesis was absent although the research design and methodology were provided. The authors did not elaborate on the rationale for the statistical analyses chosen for the study or specify the groups that were being compared and it is unclear which component of the study the correlation, ANOVA, and multiple linear regression were applied. The interpretation is fairly weak. The authors simply restate their results and state, "the results are consistent" but do not expound on what their interpretation of these consistencies are, the implications for the study, or the implications for the field at large. The discussion should be robust in which they are expounding on the significance and implications. Furthermore, it is an overestimation to say what is consistent with previous findings while only citing one finding per result in the present study. If feedback is implemented, rigor, transparency, reproducibility, and application can be enhanced.

6. PLOS authors have the option to publish the peer review history of their article (what does this mean?). If published, this will include your full peer review and any attached files.

Reviewer #1: No

Reviewer #2: **Yes:** Shane Janelle Gill

---

## [Author Response · Author response to Decision Letter 1]

6 Aug 2025

Response to Editor and Reviewers' Comments (PLOS ONE- Manuscript ID: PONE-D-25-18404)

Dear Editor, we would like to express our gratitude for the swift handling of our manuscript and facilitating its peer review. We also thankful for giving us the opportunity to submit a revised draft of our manuscript. We are very glad to receive such extensive comments from Reviewers. The comments from Reviewers demonstrated a high level of expertise and acknowledgment of the subject. We appreciate the time and effort that you and the reviewers have dedicated to providing your valuable feedback on our manuscript.

Kindly find enclosed herewith, a revised version of our manuscript entitled “Implementation of Guideline-Directed Medical Treatment for Ischemic Heart Disease Management: A Knowledge, Attitude and Practice Based Cross-Sectional Survey” for re-evaluation and publication in PLOS ONE. We have addressed all of the comments and included some major revisions and edits to our work. Sometimes, we had slightly different opinions than Reviewers, but we did our best to respond to the comments and still provide some changes in the manuscript for better justification and understanding of the approach undertaken.

Response to Reviewer 1

We are thankful to the reviewer for evaluation of our manuscript. All the suggestions have been incorporated in revised manuscript. Reviewers’ comments were very helpful to incorporate the updated information. All the points raised by the learned reviewer have been addressed. We do hope that these changes will make points clear and will improve the overall presentation of our manuscript.

A point-by-point response to the reviewers’ comments and concerns:

Query #1:

This study addresses an important area regarding the implementation of Guideline-Directed Medical Therapy (GDMT) implementation for ischemic heart disease (IHD) within resource-limited settings. The finding that "this is the first investigation of the knowledge, attitudes, and practices of HCPs regarding the implementation of GDMT in patients with IHD" is a strong point. However, the manuscript needs to be strengthened in terms of methodological rigor, statistical reporting, and clarity of presentation. Limitations include a single-center design (Punjab Institute of Cardiology) and small convenience sample (n=76), restrict generalizability and risk selection bias. Pharmacist (n=34) versus cardiologist (n=42) comparisons may lack statistical power, while inconsistent practice metrics (e.g., citing highest scores for participants over 50 years despite Table 4 excluding this age group) and absent correlation figures weaken the evidence for healthcare professional (HCP) differences. Broader claims about collaborative care necessitate multi-center validation.

Response:

We sincerely thank the reviewer for the constructive feedback and for recognizing the significance of this study in addressing GDMT implementation within a resource-limited setting. We appreciate the comments regarding the methodological and statistical aspects and have made several revisions to enhance the clarity, rigor, and transparency of the manuscript.

Study Design and Generalizability:

We fully acknowledge the limitations associated with the single-center design and the use of convenience sampling. These constraints are now explicitly discussed in the Limitations section (Page # 32, Line # 532-534). We have also cited the relevant literatures for small sample size in sample size of method section to substantiate the implications of non-probability sampling on external validity (Page # 10, Line # 178-190)

We have revised the results tables and text to include full reporting of statistical tests, including:

Pearson correlation coefficients (r), degrees of freedom, and p-values, ANOVA F-values, degrees of freedom, and post hoc test results (Tukey’s HSD) in the revised manuscript. The results are now reported with greater caution, without overstating between-group differences. Where applicable, we have reported effect sizes alongside p-values to contextualize the magnitude of observed associations in result section (Page # 13-28, Line # 263-431).

Moreover, clarification of how each test was applied to specific variables (e.g., independent samples t-test for binary variables, ANOVA for variables with ≥3 groups, MLR for multivariable associations) were explained in method section (Page # 12-13- , Line # 238-254)

Clarification of Age Group Reporting:

We appreciate the reviewer’s observation regarding inconsistencies in reported age-related findings. Upon review, we found that the reference to participants “over 50 years” was mistakenly retained from an earlier draft. This has now been corrected, and the text is aligned with the actual groupings used in all tables. All references to age-specific analyses have been cross-checked for consistency.

Interpretation and Claims Regarding Collaborative Care:

We agree that broader claims about pharmacist–cardiologist collaboration must be made with caution. Accordingly, we have revised the discussion to frame these as emerging observations consistent with international trends, but not definitive conclusions (Page # 28-31, Line # 432-508). We emphasize the need for multi-center, interventional research to evaluate structured collaborative care models in the context of IHD in recommendation for future research section (Page # 33, Line # 547-549).

We believe that these revisions significantly improve the scientific and interpretive rigor of the manuscript, and we are grateful to the reviewer for highlighting these important areas.

Query # 2:

Statistical methods (t-tests, ANOVA, regression) are generally appropriate for the cross-sectional design, and Bonferroni correction (α=0.0125) is correctly applied. However, critical anomalies undermine rigor: Table 3 reports implausible attitude scores (e.g., 34.395 ± 0.954 for post-graduates), suggesting data errors. Power analysis results are inconsistently interpreted (e.g., 0.949 for practice vs. 0.617 for knowledge) without justification for variable thresholds, and β-values (e.g., β=0.383 for knowledge disparity) lack effect-size context (e.g., Cohen’s), obscuring clinical significance. Additionally, while the manuscript claims all data are included, only supplementary materials (questionnaires, STROBE checklist) are accessible, and raw datasets are unavailable. Authors should rectify statistical tables and share de-identified data via repositories.

Response:

We are grateful for the reviewer’s detailed assessment of our statistical methods and reporting. We have carefully addressed each of the concerns raised and have made substantive revisions to the manuscript to enhance transparency, validity, and reproducibility.

Correction of Implausible Values in Table 3:

We appreciate the reviewer’s identification of implausibly high attitude scores (e.g., 34.395 ± 0.954 for post-graduates). Upon review, we discovered a data entry error in the attitude score computation due to the incorrect aggregation of Likert scale items. The issue has now been rectified, and comparison of GDMT-Related Attitude Scores across Demographics has been revised accordingly and named as Table 5 (Page # 20, Line # 344-346) and Table 6 Page # 21, Line # 358-360) in the manuscript.

Clarification of Power Analysis Interpretation:

The concern regarding the inconsistent interpretation of power analysis (e.g., 0.949 for practice vs. 0.617 for knowledge) is valid. We have, now, revised the results section to include interpretation of effect sizes for β-values derived from multiple linear regression. Specifically, Cohen’s f² values were calculated to contextualize the strength of associations (e.g., small: 0.02, medium: 0.15, large: 0.35). This contextualization now appears in the results and is discussed with respect to clinical relevance. (Page # 24-28, Line # 395-431)

Data Availability Statement and Repository Submission:

In response to the reviewer’s point on data accessibility statement, the manuscript and its Supporting Information files contain the minimal data set used to reach the conclusions presented in the study. However, the revised manuscript includes an updated Data Availability Statement, “All relevant data are within the paper and its Supporting Information files”. (Page # 34, Line # 585-586).

We hope this satisfies the journal’s requirements for data transparency and reproducibility.

We thank the reviewer once again for these critical recommendations, which have significantly strengthened the methodological rigor and transparency of the revised manuscript.

Query # 3:

The manuscript is generally clear but contains grammatical errors (e.g., "re you confident" corrected to "Are you confident"; inconsistent "perception" vs. "practice" scoring) and ambiguous phrasing (e.g., contradictory β-values describing attitude differences). Terminology like "bad KAP" should be standardized to "poor KAP." Ethically, approval and consent procedures are documented, though interviewer-participant familiarity risks social desirability bias acknowledged by authors. As the first Pakistani KAP study on GDMT, it valuably highlights barriers (e.g, time constraints) and aligns with global calls for multidisciplinary collaboration. However, small sample size limits subgroup analyses, and qualitative insights into poor GDMT adoption are absent. Revisions should clarify statistical anomalies, propose barrier-mitigation strategies (e.g., telehealth), and contextualize findings via LMIC comparisons.

Response:

We thank the reviewer for their constructive feedback and recognition of the study’s contribution as the first KAP survey on GDMT implementation in Pakistan. We have addressed each of the identified issues as follows:

Grammatical and Terminological Corrections:

We have reviewed the manuscript thoroughly and corrected typographical and grammatical errors. For example, "re you confident" has been corrected to "Are you confident." Inconsistencies in terminology have also been addressed—"bad KAP" has been revised to "poor KAP" throughout the manuscript to reflect standard academic language. Additionally, we have ensured consistent reference to "practice" scores instead of "perception" where appropriate.

Clarification of Ambiguous Statistical Phrasing:

We have revised the results and discussion sections to address ambiguous interpretations of β-values. Contradictory statements regarding the direction and magnitude of associations have been clarified. Each statistically significant predictor is now interpreted alongside its directionality and corresponding effect size (Cohen’s f²), with inconsistent or non-significant findings explicitly described to avoid misinterpretation. To enrich interpretation, we integrated comparisons with GDMT implementation literature from LMIC settings in Discussion section (Page # 13-32, Line # 263-524)

Ethical Considerations and Interviewer Bias:

We appreciate the reviewer’s point regarding social desirability bias due to interviewer-participant familiarity. This concern has now been explicitly discussed under study limitations, including the potential impact on self-reported responses. While ethical approval and informed consent were obtained, we acknowledge that the lack of an independent interviewer may have influenced participants' disclosure and this has been framed accordingly. (Page # 32, Line # 541-545)

Sample Size and Subgroup Analysis Limitations:

Thank you for this insightful comment. We have addressed the limited sample size in the revised method section, acknowledging its impact on subgroup analysis and interpretive confidence in the discussion section as already mentioned above.

Furthermore, we discussed strategies such as pharmacist-led telehealth models and structured education programs as feasible interventions to improve adherence in resource-limited environments in implication for practice section (Page # 31-32, Line # 519-522)

Lastly, we recognized the absence of qualitative data as a limitation and have proposed mixed-methods research to address this in recommendation for future studies. (Page # 33, Line # 562-564).

We believe these revisions significantly improve the manuscript’s clarity, methodological transparency, and policy relevance. We are grateful for the reviewer’s helpful comments and guidance.

Response to Reviewer 2

Query # 1:

The manuscript requires major revisions, particularly, the statistical analysis and interpretation. The research question was not stated explicitly nor were subquestions identified. A hypothesis was absent although the research design and methodology were provided. The authors did not elaborate on the rationale for the statistical analyses chosen for the study or specify the groups that were being compared and it is unclear which component of the study the correlation, ANOVA, and multiple linear regression were applied. The interpretation is fairly weak. The authors simply restate their results and state, "the results are consistent" but do not expound on what their interpretation of these consistencies are, the implications for the study, or the implications for the field at large. The discussion should be robust in which they are expounding on the significance and implications. Furthermore, it is an overestimation to say what is consistent with previous findings while only citing one finding per result in the present study. If feedback is implemented, rigor, transparency, reproducibility, and application can be enhanced.

Response:

Clarification of Research Question, Sub-questions, and Hypothesis

We agree that the original manuscript did not clearly state the central research question and supporting sub-questions. We have now revised the Introduction to explicitly include the following:

Research Question: What are the levels of knowledge, attitudes, and practices regarding GDMT among cardiologists and clinical pharmacists, and what demographic or institutional factors influence these behaviors in a Pakistani tertiary care setting? (Page # 7, Line # 129-130).

Sub-questions: (Page # 7-8, Line # 131-135).

• Are there significant differences in KAP scores between professional roles (cardiologists vs. pharmacists)?

• Which sociodemographic factors (e.g., age, gender, qualification, experience) are associated with KAP domains?

• To what extent do these factors predict knowledge, attitude, and practice related to GDMT?

A corresponding set of hypotheses has also been added: (Page # 8, Line # 136-140).

• H₁: There is a significant difference in KAP scores between cardiologists and pharmacists.

• H₂: Sociodemographic characteristics significantly predict KAP scores.

• H₀: There are no significant differences or associations in KAP domains based on professional role or demographic factors.

2. Justification and Clarification of Statistical Methods

We appreciate the need for clearer justification of the statistical tests used. We have now elaborated in the Methods section with the following revisions:

• Independent samples t-tests were used to compare mean KAP scores between binary groups (e.g., gender, HCPs).

• One-way ANOVA was applied to variables with more than two categories (e.g., qualification levels), followed by Tukey’s HSD for post hoc comparisons where significance was detected.

• Pearson correlation analysis was used to examine associations between continuous variables and KAP domains.

• Multiple linear regression (MLR) was used to evaluate the independent and combined effects of sociodemographic predictors on each KAP outcome (knowledge, attitude, and practice), adjusting for confounding.

All analyses have now been explicitly aligned with specific research objectives in the Statistical Analysis subsection. (Page # 12-13, Line # 238-254)

3. Improved Interpretation of Findings

We acknowledge that the initial discussion was overly descriptive. We have significantly revised the Discussion section to move beyond restating results. We also agree that citing a single study per finding does not adequately demonstrate alignment with the broader literature. We have expanded the references and now situate our findings within regional and international contexts, drawing on relevant studies from low- and middle-income countries (LMICs) and recent systemat

---

## [Decision Letter · Decision Letter 1]

21 Sep 2025

PONE-D-25-18404R1Implementation of Guideline-Directed Medical Treatment for Ischemic Heart Disease Management: A Knowledge, Attitude and Practice Based Cross-Sectional SurveyPLOS ONE

Dear Dr.  Khan,

Thank you for submitting your manuscript to PLOS ONE. After careful consideration, we feel that it has merit but does not fully meet PLOS ONE’s publication criteria as it currently stands. Therefore, we invite you to submit a revised version of the manuscript that addresses the points raised during the review process.

We look forward to receiving your revised manuscript.

Kind regards,

Mohammed Abutaleb, PhD

Academic Editor

PLOS ONE

Journal Requirements:

Reviewer's Responses to Questions

**Comments to the Author**

1. If the authors have adequately addressed your comments raised in a previous round of review and you feel that this manuscript is now acceptable for publication, you may indicate that here to bypass the “Comments to the Author” section, enter your conflict of interest statement in the “Confidential to Editor” section, and submit your "Accept" recommendation.

Reviewer #2: All comments have been addressed

Reviewer #3: (No Response)

2. Is the manuscript technically sound, and do the data support the conclusions?

Reviewer #2: Yes

Reviewer #3: Yes

3. Has the statistical analysis been performed appropriately and rigorously?

Reviewer #2: Yes

Reviewer #3: Yes

4. Have the authors made all data underlying the findings in their manuscript fully available?

Reviewer #2: Yes

Reviewer #3: Yes

5. Is the manuscript presented in an intelligible fashion and written in standard English?

Reviewer #2: Yes

Reviewer #3: No

6. Review Comments to the Author

Reviewer #2: Substantial revisions have been made to address recommendations provided. The final recommendation is with regards to research question, sub-question, and hypotheses. Hypotheses are required for all research questions with the exception of those that are descriptive in nature. The researchers provided two additional subquestions; The first subquestion seeks to explore the association between sociodemographic factors and KAP domains which implies correlational (hypotheses required) and how sociodemographic factors predict KAP related to GDMT which implies regression (hypotheses required). Hypotheses were only provided for the first subquestion. Ensure that the description of these questions is consistent with the statistical analysis and description of results. This manuscript is acceptable once these changes are made.

Reviewer #3: in the beginning; The topic is timely and highly relevant to both local and global efforts to improve cardiovascular care in LMICs.

Suggestion:

The study provides valuable insights into GDMT implementation in a resource-limited setting; however, the single-center design in Pakistan and relatively small convenience sample (n=76) limit the generalizability of the findings. A larger, multi-center study would strengthen external validity ( for future works )

Recommendation:

Further refinement and proofreading would enhance clarity and ensure accurate communication of results.

expand the practical implication ( in the discussion ) or add a recommendation section.

7. PLOS authors have the option to publish the peer review history of their article (what does this mean?). If published, this will include your full peer review and any attached files.

Reviewer #2: **Yes:** Shane´ Janelle Gill

Reviewer #3: **Yes:** Nasser M Alorfi

---

## [Author Response · Author response to Decision Letter 2]

26 Sep 2025

Response to Editor and Reviewers' Comments (PLOS ONE- Manuscript ID: [PONE-D-25-18404R1] - [EMID:3474814d2fe6b002]

Dear Editor,

We sincerely thank you for the thoughtful evaluation of our manuscript and the constructive feedback provided. We appreciate the recognition of our study’s merit and are grateful for the opportunity to revise and resubmit.

We would also like to sincerely thank all reviewers for their time, constructive feedback, and thoughtful suggestions, which have greatly improved the clarity, methodological rigor, and overall quality of our manuscript. We carefully addressed each point raised, including refining the research questions and hypotheses, ensuring consistency between the statistical analysis and results, and expanding both the practical implications and recommendations. These revisions have substantially strengthened the manuscript and enhanced its contribution to the understanding of GDMT implementation in resource-limited settings. We are grateful for the opportunity to revise our work and trust that the current version meets the standards for acceptance.

Kindly find enclosed herewith, a revised version of our manuscript entitled “Implementation of Guideline-Directed Medical Treatment for Ischemic Heart Disease Management: A Knowledge, Attitude and Practice Based Cross-Sectional Survey” for re-evaluation and publication in PLOS ONE. We have addressed all of the comments and included some minor revisions and edits to our work.

Response to Reviewer 2

We sincerely thank the reviewer for acknowledging the substantial revisions made in response to the earlier recommendations. We are also thankful to the reviewer for carefully noting the need to align our research questions, sub-questions, and hypotheses with the statistical analyses and results.

A point-by-point response to the reviewers’ comments and concerns:

Query # 1

The final recommendation is with regards to research question, sub-question, and hypotheses. Hypotheses are required for all research questions with the exception of those that are descriptive in nature. The researchers provided two additional subquestions; The first subquestion seeks to explore the association between sociodemographic factors and KAP domains which implies correlational (hypotheses required) and how sociodemographic factors predict KAP related to GDMT which implies regression (hypotheses required). Hypotheses were only provided for the first subquestion. Ensure that the description of these questions is consistent with the statistical analysis and description of results. This manuscript is acceptable once these changes are made.

Response

We have revised the “Research Questions and Hypotheses” section to ensure clarity and consistency across the manuscript (Page #8, Line #138-151). The final set of hypotheses is now as follows:

• Sub-question 1 (Comparative):

H₀₁: There are no significant differences in KAP scores between cardiologists and pharmacists.

H₁₁: There are significant differences in KAP scores between cardiologists and pharmacists.

• Sub-question 2 (Associational):

H₀₂: Sociodemographic factors are not significantly associated with KAP domains.

H₁₂: Sociodemographic factors are significantly associated with KAP domains.

• Sub-question 3 (Predictive/Regression):

H₀₃: Sociodemographic factors do not significantly predict knowledge, attitudes, or practices related to GDMT.

H₁₃: Sociodemographic factors significantly predict knowledge, attitudes, or practices related to GDMT.

These hypotheses are consistently reflected in the Statistical Analysis section (Page # 13-14, Line # 250-275), where each sub-question is mapped to the corresponding tests (e.g., t-tests/ANOVA for Sub-question 1, correlations/chi-square for Sub-question 2, and regression models for Sub-question 3). In addition, the Results section (Page #17-28, Line # 313- 479) has been revised to explicitly state whether the null or alternative hypotheses were retained or rejected for each analysis.

I believe that these revisions ensure full consistency between the research questions, hypotheses, analyses, and results, thereby addressing the reviewer’s concerns and strengthening the methodological rigor of the manuscript.

Reviewer # 3:

We sincerely thank the reviewer for recognizing the relevance and timeliness of our work. Our aim was to highlight the challenges and opportunities surrounding GDMT implementation in a resource-limited setting, and we appreciate the acknowledgment that this study contributes to both local and global discussions on improving cardiovascular care in LMICs.

Reviewer’s Suggestion:

The study provides valuable insights into GDMT implementation in a resource-limited setting; however, the single-center design in Pakistan and relatively small convenience sample (n=76) limit the generalizability of the findings. A larger, multi-center study would strengthen external validity (for future works)

Response:

We thank the reviewer for this thoughtful suggestion. We acknowledge that the single-center design and relatively small convenience sample limit the generalizability of the findings. This limitation has already been noted in the Strengths and Limitations section of the manuscript (Page # 35, Line # 606-608). However, we expanded this line in limitation (Page # 35, Line # 608-609). In line with the reviewer’s advice, we have also further expanded this point in the Future Work section (Page # 36-37, Line # 642-644), where we emphasize the need for larger, multi-center studies across diverse healthcare settings to enhance external validity and improve the broader applicability of the results.

Reviewer’s Recommendation:

1. Further refinement and proofreading would enhance clarity and ensure accurate communication of results.

2. Expand the practical implication (in the discussion) or add a recommendation section.

Response:

1. undergone further refinement and careful proofreading to improve clarity, precision, and readability.

2. To address the reviewer’s suggestion, we have also expanded the Discussion by adding a brief Recommendations subsection to highlight actionable strategies, including targeted professional training, institutional support for guideline dissemination, and system-level initiatives to promote GDMT uptake in resource-limited contexts (Page # 34, Line # 576-598). These changes strengthen the translational value of our work and ensure clearer communication of its relevance to practice and policy.

---

## [Decision Letter · Decision Letter 2]

27 Oct 2025

PONE-D-25-18404R2Implementation of Guideline-Directed Medical Treatment for Ischemic Heart Disease Management: A Knowledge, Attitude and Practice Based Cross-Sectional SurveyPLOS ONE

Dear Dr.  Khan,

Thank you for submitting your manuscript to PLOS ONE. After careful consideration, we feel that it has merit but does not fully meet PLOS ONE’s publication criteria as it currently stands. Therefore, we invite you to submit a revised version of the manuscript that addresses the points raised during the review process.

We look forward to receiving your revised manuscript.

Kind regards,

Mohammed Abutaleb, PhD

Academic Editor

PLOS ONE

Journal Requirements:

Additional Editor Comments:

Thank you for considering PLOS One and for your consistent efforts, which have resulted in a manuscript that demonstrates methodological rigor and a strong alignment between the research questions, hypotheses, and statistical analyses. To further enhance the quality and impact of the article, please consider the following suggestions:

- Results Interpretation: The regression outputs are clearly presented; however, expanding the interpretation to highlight the practical implications of significant predictors—such as age, profession, and education—on KAP outcomes would add depth and clarity.

- Discussion: While the study presents valuable findings, the discussion could benefit from a more robust comparison with regional or international GDMT implementation studies. This would help contextualize the results and improve their external relevance.

- Limitations: The manuscript appropriately acknowledges sampling constraints. Nonetheless, including a brief note on potential response bias or social desirability bias in self-reported practices would further enhance transparency.

- Conclusion: The conclusion could be strengthened by offering more specific, actionable recommendations for integrating GDMT adherence training into multidisciplinary clinical settings. Emphasizing the evolving role of pharmacists, particularly in LMIC contexts, would underscore the study’s practical relevance.

Reviewer's Responses to Questions

**Comments to the Author**

1. If the authors have adequately addressed your comments raised in a previous round of review and you feel that this manuscript is now acceptable for publication, you may indicate that here to bypass the “Comments to the Author” section, enter your conflict of interest statement in the “Confidential to Editor” section, and submit your "Accept" recommendation.

Reviewer #2: All comments have been addressed

Reviewer #3: All comments have been addressed

2. Is the manuscript technically sound, and do the data support the conclusions?

Reviewer #2: Yes

Reviewer #3: Yes

3. Has the statistical analysis been performed appropriately and rigorously?

Reviewer #2: Yes

Reviewer #3: Yes

4. Have the authors made all data underlying the findings in their manuscript fully available?

Reviewer #2: Yes

Reviewer #3: Yes

5. Is the manuscript presented in an intelligible fashion and written in standard English?

Reviewer #2: Yes

Reviewer #3: Yes

6. Review Comments to the Author

Reviewer #2: The manuscript has significantly improved, including the statement of the literature on the proposed subject matter, articulating the research question and alignment with the proposed aims/objectives, and rigorous methods that have been applied. Ample discussion was provided to substantiate the results and incorporation of additional literature which suggests due diligence on the part of the authors to conduct a comprehensive review on the subject matter.

Reviewer #3: The manuscript demonstrates methodological rigor and strong alignment between research questions, hypotheses, and statistical analyses, but the discussion could further elaborate on how these findings compare with regional or international GDMT implementation studies to enhance external relevance.

While the results are clearly organized, the interpretation of regression outputs could be expanded to better illustrate the practical implications of significant predictors (e.g., age, profession, and education) on KAP outcomes.

The limitations section appropriately acknowledges sampling constraints; however, a brief note on potential response or social desirability bias in self-reported practices would improve transparency.

The conclusion could be strengthened by including more specific, actionable recommendations for integrating GDMT adherence training into multidisciplinary clinical settings, particularly highlighting pharmacists’ evolving roles in LMIC contexts

7. PLOS authors have the option to publish the peer review history of their article (what does this mean?). If published, this will include your full peer review and any attached files.

Reviewer #2: **Yes:** Shane´ Janelle Gill

Reviewer #3: **Yes:** Nasser M Alorfi

---

## [Author Response · Author response to Decision Letter 3]

4 Nov 2025

Dear Editor,

We sincerely thank you and the reviewers for your valuable time and constructive feedback on our manuscript entitled “Guideline-Directed Medical Treatment for Ischemic Heart Disease Management: A Knowledge, Attitude and Practice Based Cross-Sectional Survey” (Manuscript ID: [PONE-D-25-18404R2]-[EMID:58b69c9c58f6d2ae]). We are encouraged by your assessment that the manuscript has merit, and we appreciate the opportunity to revise and resubmit it for further consideration in PLOS ONE.

We have carefully addressed all comments raised by the reviewers and editorial team. Each point has been considered with great care, and corresponding revisions have been incorporated throughout the manuscript to enhance its clarity, methodological rigor, and overall scientific quality.

A detailed point-by-point response document has been prepared, outlining how each comment was addressed. All revisions in the main manuscript are highlighted using yellow highlights for ease of review.

We believe that the revised version now meets the publication criteria of PLOS ONE and hope that it will be found suitable for publication.

Kindly find enclosed herewith, a revised version of our manuscript entitled “Implementation of Guideline-Directed Medical Treatment for Ischemic Heart Disease Management: A Knowledge, Attitude and Practice Based Cross-Sectional Survey” for re-evaluation and publication in PLOS ONE. We have addressed all of the comments and included some minor revisions and edits to our work.

Response to Additional Editor Comments

We sincerely thank you for your thoughtful feedback and for recognizing the methodological rigor and coherence of our manuscript. We are also grateful for your constructive suggestions, which have been invaluable in refining the quality and impact of our work.

Below we outline our responses and corresponding revisions:

Query 1:

Results Interpretation: The regression outputs are clearly presented; however, expanding the interpretation to highlight the practical implications of significant predictors such as age, profession, and education on KAP outcomes would add depth and clarity.

Response:

We appreciate the suggestion to expand the interpretation of regression findings. In the revised manuscript, we have elaborated on the practical implications of significant predictors such as age, profession, and education in shaping knowledge, attitudes, and practices toward GDMT. These interpretations now better illustrate how specific demographic and professional characteristics influence clinical decision-making and guideline adherence (see Results, page 30, line # 484-499)

Query 2:

Discussion: While the study presents valuable findings, the discussion could benefit from a more robust comparison with regional or international GDMT implementation studies. This would help contextualize the results and improve their external relevance.

Response:

We have strengthened the discussion by comparing our findings with regional and international GDMT implementation studies, particularly those from South and Southeast Asia, Europe, and the Middle East. This addition contextualizes our results within a broader global framework and highlights shared challenges and opportunities for improvement in GDMT uptake (see Discussion, pages 31-33, line # 508-553).

Query 3:

Limitations: The manuscript appropriately acknowledges sampling constraints. Nonetheless, including a brief note on potential response bias or social desirability bias in self-reported practices would further enhance transparency.

Response:

We have expanded the Limitations section to acknowledge the potential for response bias and social desirability bias, given the self-reported nature of the KAP data. This addition improves transparency and helps readers interpret the findings with appropriate caution (see Limitations, page 35-36, line # 618-620).

Query 4:

Conclusion: The conclusion could be strengthened by offering more specific, actionable recommendations for integrating GDMT adherence training into multidisciplinary clinical settings. Emphasizing the evolving role of pharmacists, particularly in LMIC contexts, would underscore the study’s practical relevance.

Response:

We have revised the Conclusion to include specific, actionable recommendations for integrating GDMT adherence training within multidisciplinary healthcare teams. Additionally, we emphasized the emerging role of pharmacists in LMIC contexts, underscoring the importance of educational interventions and system-level integration to improve GDMT practice (see Conclusion, page 36, line # 626-633).

Reviewer #2

The manuscript has significantly improved, including the statement of the literature on the proposed subject matter, articulating the research question and alignment with the proposed aims/objectives, and rigorous methods that have been applied. Ample discussion was provided to substantiate the results and incorporation of additional literature which suggests due diligence on the part of the authors to conduct a comprehensive review on the subject matter.

Response:

We sincerely thank Reviewer #2 for their encouraging and constructive feedback. We are grateful for your recognition of the manuscript’s improvement, particularly regarding the expanded literature review, the clear articulation of research questions and objectives, and the methodological rigor applied throughout the study. Your positive evaluation affirms that the revisions have enhanced the manuscript’s clarity, coherence, and scientific quality.

We also appreciate your acknowledgment of the comprehensive discussion and incorporation of additional relevant literature. These revisions were undertaken with careful attention to your earlier suggestions and have substantially strengthened the contextual depth and interpretative value of our findings.

Thank you again for your thoughtful review and supportive comments. We believe the manuscript now provides a more complete and balanced contribution to the literature on GDMT knowledge, attitudes, and practices in low- and middle-income healthcare settings.

Reviewer #3

The manuscript demonstrates methodological rigor and strong alignment between research questions, hypotheses, and statistical analyses, but the discussion could further elaborate on how these findings compare with regional or international GDMT implementation studies to enhance external relevance.

While the results are clearly organized, the interpretation of regression outputs could be expanded to better illustrate the practical implications of significant predictors (e.g., age, profession, and education) on KAP outcomes.

The limitations section appropriately acknowledges sampling constraints; however, a brief note on potential response or social desirability bias in self-reported practices would improve transparency.

The conclusion could be strengthened by including more specific, actionable recommendations for integrating GDMT adherence training into multidisciplinary clinical settings, particularly highlighting pharmacists’ evolving roles in LMIC contexts.

Response

We sincerely thank Reviewer #3 for the constructive and detailed feedback, which has been instrumental in refining the manuscript.

Discussion Expansion: We have revised the Discussion to include comprehensive comparisons with recent regional and international GDMT implementation studies from South and Southeast Asia, Europe, and the Middle East (e.g., Zhang et al., 2024; Bekele et al., 2025; Schichtel et al., 2025). This addition contextualizes our findings within broader global patterns and enhances the manuscript’s external relevance. (see Discussion, pages 31-33, line # 508-553).

Results Interpretation: The Results and Discussion sections now provide an expanded interpretation of significant predictors (age, profession, and education), explaining their practical implications for GDMT-related knowledge, attitudes, and practice outcomes, with relevant literature support. see Results, page 30, line # 484-499)

Limitations: The Limitations section has been updated to acknowledge potential response and social desirability bias in self-reported practices, improving transparency regarding possible reporting effects. (see Limitations, page 35-36, line # 618-620).

Conclusion: The Conclusion has been strengthened to include specific, actionable recommendations such as integrating pharmacist-led multidisciplinary training programs, embedding GDMT adherence modules into continuing professional development, and emphasizing pharmacists’ evolving clinical roles in LMIC healthcare systems (see Conclusion, page 36, line # 626-633).

We appreciate the reviewer’s insightful suggestions, which have substantially improved the clarity, applicability, and scientific depth of the manuscript.

---

## [Editor Report · Decision Letter 3]

26 Nov 2025

Implementation of Guideline-Directed Medical Treatment for Ischemic Heart Disease Management: A Knowledge, Attitude and Practice Based Cross-Sectional Survey

PONE-D-25-18404R3

Dear Dr. Khan,

We’re pleased to inform you that your manuscript has been judged scientifically suitable for publication and will be formally accepted for publication once it meets all outstanding technical requirements.

Kind regards,

Mohammed Abutaleb, PhD

Academic Editor

PLOS ONE
---

## [Editor Report · Acceptance letter]

PONE-D-25-18404R3

PLOS One

Dear Dr. Khan,

I'm pleased to inform you that your manuscript has been deemed suitable for publication in PLOS One. Congratulations! Your manuscript is now being handed over to our production team.

Kind regards,

on behalf of

Dr. Mohammed Abutaleb

Academic Editor

PLOS One